# Recurrent Shoulder Instability after Arthroscopic Bankart Repair in an Elite Baseball Pitcher—A Case Report

**DOI:** 10.3390/medicina58111635

**Published:** 2022-11-12

**Authors:** Jo-Ting Kao, Cheng-Pang Yang, Huan Sheu, Hao-Che Tang, Yi-Sheng Chan, Alvin Chao-Yu Chen, Kuo-Yao Hsu, Chun-Jui Weng, You-Hung Cheng, Chih-Hao Chiu

**Affiliations:** 1Department of Orthopedic Surgery, Linkou Chang Gung Memorial Hospital, Taoyuan 333, Taiwan; 2Department of Orthopedic Surgery, Taoyuan Chang Gung Memorial Hospital, Taoyuan 333, Taiwan; 3Department of Orthopedic Surgery, Keelung Chang Gung Memorial Hospital, Keelung 204, Taiwan; 4Department of Orthopedic Surgery, New Taipei Municipal Tucheng Hospital, New Taipei City 236, Taiwan; 5Department of Orthopedic Surgery, Kaohsiung Municipal Feng-Shan Hospital, Kaohsiung 833, Taiwan

**Keywords:** shoulder instability, Bankart, Remplissage, Latarjet, pither

## Abstract

A 16-year-old right-handed male pitcher had a first-time right anterior shoulder dislocation during a baseball game. X-ray and MRI revealed no apparent glenoid bone loss or Hill-Sachs lesion, but an anterior labroligamentous periosteal sleeve avulsion (ALPSA) lesion with mild posterolateral decompression of the humerus head. His instability severity index score (ISIS score) was 5 with an on-track lesion. He had an arthroscopic Bankart repair using two all-suture anchors and returned to pitching 6 months after the index surgery. However, he had an unstable sensation after 50 pitches during a game one year postoperatively. This time, he presented with a significant Hill–Sachs lesion and a recurrent APLSA lesion. His ISIS score was 6 with an on-tract lesion. During the arthroscopic examination, the previous suture was stable, while anterior capsuloligament tissues were dislodged from sutures, and a Hill–Sachs lesion was observed. This time, a revision arthroscopic Bankart repair and Remplissage procedure were done on him with four double-loaded soft tissue anchors. Pitchers often develop more external rotation in their throwing arm because of a repetitive stretch of the anterior shoulder capsule and ligaments during pitching. The decrease in external rotation after surgery may limit the pitching speed of the pitcher, making a return to play (RTP) more difficult. There is still a paucity of best evidence to revise a failed arthroscopic Bankart repair in the dominant arm of a pitcher. Arthroscopic Bankart repair and Remplissage procedure have gained increasing popularity because they can provide a stable shoulder without harvesting the coracoid. The Latarjet procedure provides a high RTP rate; however, we did not perform it in the revision surgery and decided to revise the Bankart lesion again on its own with a Remplissage procedure, even with his ISIS score being 6 before the revision surgery. A salvage Latarjet procedure is left as a bailout procedure.

## 1. Introduction

Traumatic anterior shoulder instability is less common in baseball players than in players in other contact sports (football, hockey, and wrestling). It happens when a position player collides with another player, dives for a ball, or slides into a base with an outstretched arm [1]. Throwers, particularly pitchers, rely on a balance of shoulder mobility and stability to excel at their sports performance. Pitchers often develop more external rotation in their throwing arm because of a repetitive stretch of the anterior shoulder capsule and ligaments during pitching, especially those who throw hard with high pitching velocities [1]. The early cocking phase of pitching with the arm at approximately 90 degrees of abduction, 30 degrees of horizontal abduction, and 50 degrees of external rotation mimics [2] patients with a Bankart lesion complaining of instability at 90 degrees of abduction and higher degrees of external rotation, the so-called apprehension test [3]. If an anterior shoulder instability after shoulder dislocation occurs in pitchers, the treatment will become more challenging, considering that the balance of mobility and stability is difficult to preserve at an optimum state. The question of which anatomical structure, such as the anteroinferior labrum or Hill–Sachs lesion, should be repaired or augmented with a coracoid bone graft such as the Latarjet procedure [4] has remained elusive because potential stiffness after surgery may hinder a good range of motion (ROM) of the treating shoulder. Regarding elite and professional baseball players, Park et al. reported that only 64% return to play and 57% solidly return to play as pitchers following arthroscopic treatment for shoulder instability [5]. Here, we presented a case of a young pitcher who had a failed arthroscopic Bankart repair one year after surgery, which was then revised with a revision Bankart repair and a Remplissage procedure. 

## 2. Clinical Cases Presentation 

A 16-year-old right-handed male pitcher had a first-time right anterior shoulder dislocation during sliding into second base. At that time, the shoulder was reduced spontaneously. His maximum speed was up to 130 Km/hr before the dislocation but dropped to 100 Km/hr after the reduction because of the pain. He had experienced adequately periscapular muscle training, three platelet-rich plasma (PRP) injections, and one hyaluronic acid injection without improvement in 3 months. 

The thorough physical examination of his shoulder revealed full ROM, normal cuff strength, no evidence of scapular dyskinesis, and a completely normal neurovascular exam without hyperlaxity. On the affected throwing right arm, he demonstrated a positive active compression test [6], biceps load test [7], and apprehension test at maximum abduction and external rotation [3]. 

Radiographs of the right shoulder revealed no apparent glenoid bone loss or Hill–Sachs lesion (Figure 1A,B). MRI showed a right anterior labroligamentous periosteal sleeve avulsion (ALPSA) lesion with mild posterolateral decompression of the humerus head (Figure 1C,D). No obvious glenoid and humeral bone loss was observed, as it was classified as an on-tract lesion according to Di Giacomo et al. [8]. His instability severity index score (ISIS score) was 5 [9].

Due to his inability to return to play in elite baseball games after 3 months of conservative treatment, the patient was brought to the operation room for arthroscopic soft tissue stabilization. He was put in a beach chair position with an interscalene nerve block. The ALPSA lesion was fixed with four simple sutures from two all-suture anchors(Y-Knot Flex, ConMed Linvatec, Largo, FL, USA) (Figure 2A,B). After the surgery, the operated arm was immobilized using a sling for the first 2 weeks. The patient was instructed to perform self-rehabilitation exercises at home, starting on the first postoperative day, following a protocol designed by Roulet et al. [10]. A return to low-risk sports was allowed at 6 weeks after surgery and a return to throwing at 3 months postoperatively. He regained ROM of the right shoulder 4 months postoperatively and went back to full baseball training at 6 months postoperatively. He could pitch in bullpen and regular baseball games without difficulty 9 months after the surgery with a ROWE score [11] of 100 points and a 95% subjective shoulder value [12].

However, he had an unstable sensation after 50 pitches during a game one year after the surgery. He presented to the physician’s office again, whereupon examination showed a positive apprehension test [3] of the right shoulder. Radiographs and MRI revealed a significant Hill–Sachs lesion and a recurrent APLSA lesion (Figure 3A–D). His ISIS score was 6 this time [9] with an on-tract lesion. During the arthroscopic examination, the previous suture was stable, while anterior capsuloligament tissues were dislodged from sutures (Figure 3E). A Hill–Sachs lesion was observed (Figure 3F). This time, a revision arthroscopic Bankart repair and Remplissage procedure were done on him with 4 double-loaded soft tissue anchors (Iconix, Stryker, Mahwah, NJ, USA). One was used for the Remplissage procedure with two mattress sutures, and the other three were used for revision Bankart repair with six simple sutures (Figure 3G,H).

## 3. Discussion

### Anterior Shoulder Instability and the Associated Lesions

Bankart lesion, with a detachment of the anterior labrum from the glenoid rim, is a common cause of anterior shoulder instability in athletes after shoulder dislocation [13,14]. The other common injuries include anterior glenoid fracture, bony Bankart, and ALPSA lesion [1,15,16]. Shoulder dislocation can also result in a Hill–Sachs lesion, with a posterior humeral head impaction caused by the anterior glenoid when the humeral head dislocates anteriorly. Treatments for anterior shoulder instability include arthroscopic soft tissue repair or bone block procedures (including iliac crest or distal tibial allografts), Latarjet procedures, and auxiliary procedures including Remplissage procedures [16].

### Bankart Repair

Arthroscopic Bankart repair is most often used for anterior shoulder instability in throwers, and it repairs the labrum back down to the glenoid rim with suture anchors to tighten the capsule and re-tension the glenohumeral ligaments [1]. Although both open and arthroscopic Bankart repair showed similar outcomes [17], arthroscopic Bankart repair led the trend due to a better recovery rate for external rotation, regardless of higher recurrence and reoperation rates than open Bankart repair [18].

### Return to Play after Bankart Repair

Arthroscopic Bankart repair provides 78% to 98% return to play (RTP) rates [19,20,21,22,23]. However, the RTP of athletes depends on the sport types, the level of competition, and the demands of the affected shoulder in the associated sports. Non-contact, non-throwing athletes had the best outcome after Bankart repair (90~100% RTP) [24]. Collision and contact athletes also showed good outcomes with 90% RTP [24,25]. Throwers had the worse outcome, with 60~80% RTP, compared to that of other athletes [24]. The RTP among baseball players is even more different than in other sports, depending on the position they played. Park et al. evaluated the clinical outcome of arthroscopic Bankart repair for anterior shoulder instability in 51 elite and professional baseball players. They found that players with the injured non-throwing shoulder or infielders yielded the best results. Catchers had 83% RTP, and infielders had 90% RTP. The difference in RTP between injured non-throwing and throwing shoulders showed no significant differences in catchers and infielders. However, pitchers with an injured throwing shoulder only had 20% RTP and 0% solid RTP, compared to 89% RTP and solid RTP of pitchers with injuries in the non-throwing shoulder [5].

### Latarjet Procedure

Although primary arthroscopic Bankart repair for anterior shoulder instability yielded good outcomes [24], even for overhead athletes [24,26], some surgeons prefer the Latarjet procedure for the initial treatment of shoulder instabilities, even in patients without notable bone loss [27,28]. It results in excellent functional outcomes, even in the long term [29], and provides a low rate of recurrent instability, a high RTP, and a high rate of patient satisfaction [30,31]. Currently, the primary Latarjet procedure is considered for patients with a high risk of instability or recurrence, such as young and active individuals who participate in overhead and contact sports [16]. The Latarjet procedure provided 73~88% overall RTP for athletes [29,32], with 88.2% RTP in collision athletes and 90.3% RTP in overhead athletes [33]. Despite the high RTP for athletes, it remains controversial in throwers due to the following reasons. First, there is the possibility of losing postoperative external rotation after the Latarjet procedure [30], which is not tolerable for throwers, particularly for pitchers. Second, an average of 4.5% glenoid bone defect was reported in throwers after traumatic anterior instability, lower than the 8.9% glenoid bone loss in American football players and 12% in rugby players. Among throwers, 70% have no glenoid bone defect, and only 7% have more than a 20% glenoid bone defect [30,34]. While the percentage of critical glenoid bone loss is highly debated considering on-track or off-track Hill–Sachs lesion, it is generally accepted that it would be better to perform the Latarjet procedure in patients with 13.5% to 20% glenoid bone defect, which is far higher than average bone defects in throwers [16]. Therefore, although the literature showed similar outcomes for Bankart repair and Latarjet procedure for anterior should instability, the Latarjet procedure is not commonly accepted as a first-line treatment for throwers with shoulder instabilities, especially for pitchers, due to the potential loss of external rotation and a higher complication and revision rate [35].

### Failed Bankart Repair

Although arthroscopic Bankart repair has gained popularity for treating shoulder instability, residual instability was still reported after such a procedure. The risk factors of recurrent shoulder instability are multifactorial, including young age, male sex, an increased number of preoperative dislocations, competitive sports, duration of symptoms, glenoid bone loss, and Hill–Sachs lesions [36,37]. Many studies reported that young age was an independent risk factor for recurrent instability following Bankart repair [36,37], and US epidemiologic data also showed that patients between 16 to 20 years old demonstrated the highest rates of recurrence after surgery, up to 24.8% [38]. When treating recurrent shoulder instability after Bankart repair, the identification of the specific cause of failure is very important, especially glenoid or humeral head bone loss. If there is subcritical glenoid bone loss with non-engaging Hill–Sach lesions, open or arthroscopic Bankart revision repair can be applied. The Latarjet procedure should be left for competitive or contact athletes with a failed previous Bankart repair [39]. For a subcritical glenoid bone with engaging Hill–Sach lesions, revision Bankart repair with a Remplissage procedure or Latarjet procedure is the treatment option. If there is a critical glenoid bone loss comprising 20% to 25% or more of the inferior glenoid diameter, bone block procedures such as the Latarjet procedure, iliac crest autograft, or distal tibial allograft augmentation must be addressed [16].

### Salvages for Failed Bankart Repair

Arthroscopic Bankart repair and a Remplissage procedure can re-establish stability in patients with subcritical glenoid bone loss and engaging Hill–Sach lesions with less than a 10% recurrence rate [40,41,42]. They have gained increasing popularity and has even been considered for non-engaging Hill–Sachs lesions in professional collision athletes [43]. However, there are still concerns about postoperative stiffness and external rotation loss in arthroscopic Bankart repair with the Remplissage procedure. A systematic review showed a clear decrease in external rotation in most biomechanical studies, whereas there was only a small decrease in clinical studies [44]. However, the subclinical loss of external rotation may have negative implications for throwing mechanics. Hence, it is not an optimal primary treatment for anterior shoulder instability without a Hill–Sachs lesion in throwers, especially for pitchers. There were very few studies focusing on pitchers who received arthroscopic Bankart repair with the Remplissage procedure. One case report presented a 24-year-old triple-A minor league pitcher who had a right-shoulder first-time dislocation when sliding into second base. He had an arthroscopic Bankart repair with a Remplissage procedure for an anterior labral tear and Hill–Sachs lesion, with a subtle anterior glenoid bone loss [45]. Although the Remplissage procedure has not been associated with any appreciable loss of motion, which qualifies it as a reasonable option for pitchers, a diminished velocity may be a critical concern due to a loss of external rotation after such a procedure. On the other hand, although the salvage Latarjet procedure had high RTP rates of up to 91% [46], the significantly decreased loss of external rotation may impair the career of throwers.

Balg et al. developed an ISIS score and found that patients with an ISIS score of 3–6 had a 9.9% failure rate following primary arthroscopic Bankart repair. A Latarjet procedure is suggested for these patients. However, due to the prerequisite of good ROM in the pitching arm of the treated shoulder, we did not perform the Latarjet procedure in the revision surgery and decided to revise the Bankart lesion again on its own with a Remplissage procedure, even with his ISIS score being 6 before the revision surgery. There is still a paucity of best evidence on revising a failed arthroscopic Bankart repair in the dominant arm of a pitcher. A salvage Latarjet procedure has been left as a bailout procedure in case the revision Bankart repair and Remplissage procedure still fail in this patient.

## 4. Conclusions

Pitchers rely on the balance of shoulder mobility and stability to excel at their performance. The ROM of the pitching shoulder, especially external rotation, is especially important for pitchers. Arthroscopic Bankart repair with a Remplissage procedure is a reasonable option for pitchers with anterior shoulder instability with Hill–Sachs lesions. A salvage Latarjet procedure is left as a bailout procedure.

## Figures and Tables

**Figure 1 medicina-58-01635-f001:**
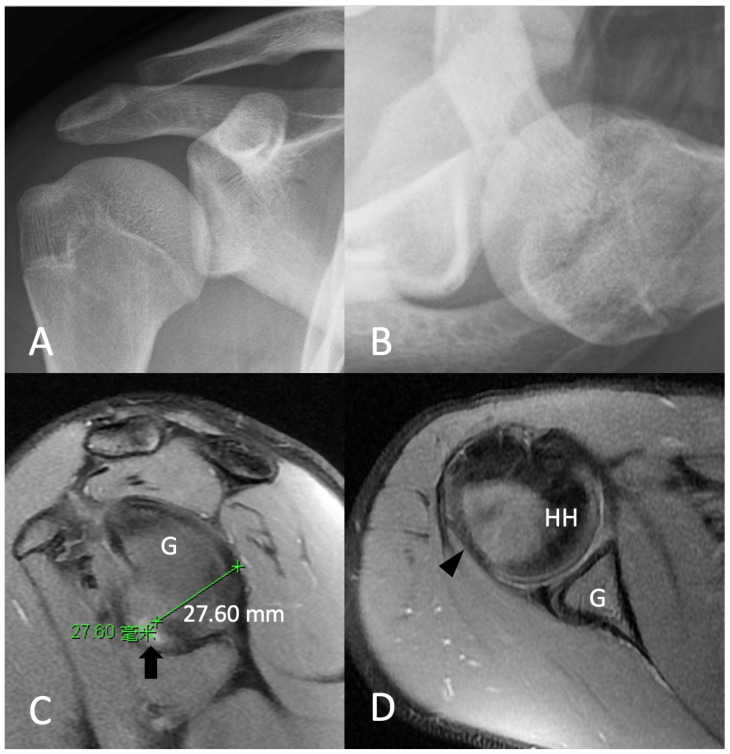
Preoperative X-ray and MRI of right shoulder revealed (**A**,**B**) no apparent glenoid bony lesion or Hill–Sachs lesion. (**C**) Anterior labroligamentous periosteal sleeve avulsion (arrow) and (**D**) mild posterolateral decompression of humerus head (arrowhead). G, glenoid; HH, humeral head.

**Figure 2 medicina-58-01635-f002:**
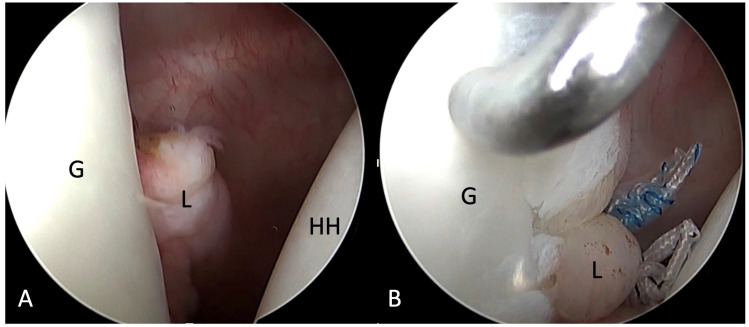
(**A**) Right anterior labroligamentous periosteal sleeve avulsion lesion (**B**) repaired with two double-loaded all-suture anchors. G, glenoid; HH, humeral head; L, labrum.

**Figure 3 medicina-58-01635-f003:**
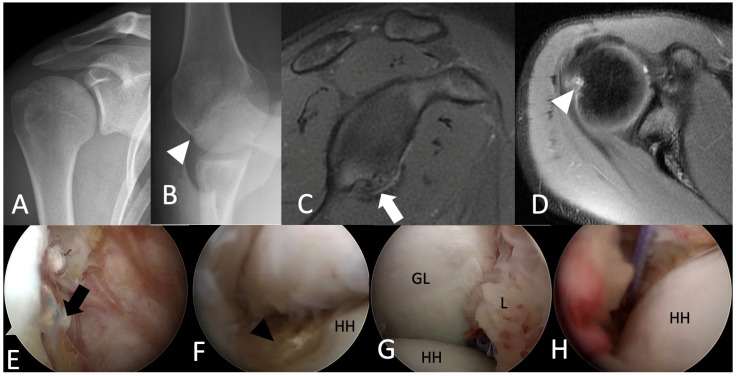
(**A**,**B**) Radiographs revealed a significant Hill–Sachs lesion (white arrowhead). (**C**) MRI revealed a recurrent ALPSA lesion (white arrow) and (**D**) a Hill–Sachs lesion (white arrowhead). (**E**) The previous suture was stable in place (black arrow), while anterior capsuloligament tissues were dislodged from sutures. (**F**) A Hill–Sachs lesion (black arrowhead). (**G**) A revision arthroscopic Bankart repair and a (**H**) Remplissage procedure were done. HH, humeral head. GL, glenoid; L, labrum.

## Data Availability

Not applicable.

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
