# Peer review of "Recurrent Shoulder Instability after Arthroscopic Bankart Repair in an Elite Baseball Pitcher—A Case Report"

_medicina, 2022, doi:10.3390/medicina58111635_

Round 1

Reviewer 1 Report

The article describes the treatment of a pincher's shoulder instability as well as the management of the treatment failure.

Sports-related injuries in the upper limb certainly have less space in the literature than those of the lower limb. Such a case, especially considering the need for revision surgery, provides important elements to help the surgeon in making choices in similar situations.

The article is very well written in my opinion: the case clearly presented. In particular, I found the discussion regarding the advantages and disadvantages of choosing between the Bankart repair or the Latarjet procedure, either as the first treatment or as the treatment of recurrence, very interesting.

I think some parts can be improved before publication, in particular:

1) the introduction is too vague, it should give some more information about the pathology and mention the treatment options; also some information about the size of the problem from the epidemiological point of view;

2) the iconography I think could be perfected: it would be helpful if the pictures were included in the text with larger size and more anatomical details indicated with pointers and made explicit in the figure legends.

Thank you.

Author Response

Reviewer 1

  • the introduction is too vague, it should give some more information about the pathology and mention the treatment options; also some information about the size of the problem from the epidemiological point of view;

Reply:

The pathology why overhead pitching leads to anterior shoulder instability is added into the revised manuscript as “The early cocking phase of pitching with the arm at approximately 90 degrees of abduction, 30 degrees of horizontal abduction, and 50 degrees of external rotation mimics[1] mimics the patients with Bankart lesion complaining of instability at 90 degrees of abduction and higher degrees of external rotation, the so-called apprehension test.[2]”.

The information about the treatment options is revised as follows “”Which anatomical structure such as the anteroinferior labrum or Hill-Sachs lesion, should be repaired or augmented with a coracoid bone graft such as the Latarjet procedure, [3] remained elusive because the potential stiffness after the surgery may hinder the good range of motion (ROM) of the treating shoulder.”

The information about the size of the problem from the epidemiological point of view is revised as follows, “Regarding elite and professional baseball players, Park et al. reported only 64% return to play and 57% solidly return to play in pitchers following arthroscopic treatment for shoulder instability.[4]”. Thanks for the comment

  • the iconography I think could be perfected: it would be helpful if the pictures were included in the text with larger size and more anatomical details indicated with pointers and made explicit in the figure legends.

Reply:

Figure 1 was revised as follows, “Figure 1. Preoperative X-ray and MRI of right shoulder revealed, (A, B) no apparent glenoid bony lesion or Hill-Sachs lesion. (C) Anterior labroligamentous periosteal sleeve avulsion (arrow), and (D) mild posterolateral decompression of humerus head (arrowhead). G, glenoid; H, humerus.”

Figure 2 was revised as follows, “Figure 2. (A) Right anterior labroligamentous periosteal sleeve avulsion lesion. (B) Repaired with two double-loaded all-suture anchors. G, glenoid; HH, humeral head; L, labrum.”

Figure 3 was revised as follows, “Figure 3. (A, B) Radiographs revealed a significant Hill-Sachs lesion (white arrowhead). (C) MRI revealed a recurrent ALPSA lesion (white arrow), and (D) a Hill-Sachs lesion (white arrowhead). (E)The previous suture was stable in place (black arrow), while anterior capsuloligament tissues were dislodged from sutures. (F) A Hill-Saches lesion (black arrowhead). (G) A revision arthroscopic Bankart repair and, a (H) Remplissage procedure were done. HH, humeral head. GL, glenoid; L, labrum.”

Thanks for the comment

References

  1. Park, S.S.; Loebenberg, M.L.; Rokito, A.S.; Zuckerman, J.D. The shoulder in baseball pitching: biomechanics and related injuries--Part 1. Bulletin of the NYU Hospital for Joint Diseases 2002, 61, 68-68.
  2. Skendzel, J.G.; Sekiya, J.K. Diagnosis and management of humeral head bone loss in shoulder instability. Am J Sports Med 2012, 40, 2633-2644, doi:10.1177/0363546512437314.
  3. Latarjet, M. [Treatment of recurrent dislocation of the shoulder]. Lyon Chir 1954, 49, 994-997.
  4. Park, J.-Y.; Lee, J.-H.; Oh, K.-S.; Chung, S.W.; Lim, J.-j.; Noh, Y.M. Return to play after arthroscopic treatment for shoulder instability in elite and professional baseball players. Journal of Shoulder and Elbow Surgery 2019, 28, 77-81.
  5. Rowe, C.R.; Zarins, B. Chronic unreduced dislocations of the shoulder. J Bone Joint Surg Am 1982, 64, 494-505.
  6. Williams, G.N.; Gangel, T.J.; Arciero, R.A.; Uhorchak, J.M.; Taylor, D.C. Comparison of the Single Assessment Numeric Evaluation method and two shoulder rating scales. Outcomes measures after shoulder surgery. Am J Sports Med 1999, 27, 214-221.

Reviewer 2 Report

This case study was clearly presented and the findings further contributes to the evidence (Ialenti et al, 2017 & Brown, et al, 2017), on the use of Bankart repair and Remplissage procedure for the management of significant Hill-Sachs lesion and a recurrent APLSA lesion.

The methods adopted by the author (s) are appropriate in addressing the objective of the case and the findings interpreted appropriately including the discussion. However, the authors did not used validated outcome measures to describe functional improvements at 6 weeks, 3, 4, 6 and 9 months.

A small typo error is noted in line 107 of Figure 3. (A, B) ….recurrent APLSA (authors to change to ALPSA) lesion (white arrow),….

References

Ialenti, M. N., Mulvihill, J. D., Feinstein, M., Zhang, A. L., & Feeley, B. T. (2017). Return to Play Following Shoulder Stabilization: A Systematic Review and Meta-analysis. Orthopaedic journal of sports medicine, 5(9), 2325967117726055. https://doi.org/10.1177/2325967117726055.

Brown, L., Rothermel, S., Joshi, R., & Dhawan, A. (2017). Recurrent Instability After Arthroscopic Bankart Reconstruction: A Systematic Review of Surgical Technical Factors. Arthroscopy: the journal of arthroscopic & related surgery: official publication of the Arthroscopy Association of North America and the International Arthroscopy Association, 33(11), 2081–2092. https://doi.org/10.1016/j.arthro.2017.06.038.

Author Response

Reviewer 2

The authors did not used validated outcome measures to describe functional improvements at 6 weeks, 3, 4, 6 and 9 months.

Reply: Thanks for the comment. We only check his outcome 9 moths after the primary surgery with a ROWE score of 100 points and 95% subjective shoulder value. Therefore, the manuscript was revised as “He could pitch in bullpen and regular baseball games without difficulty 9 months after the surgery with a ROWE score[5] of 100 points and 95% subjective shoulder value.[6]”

A small typo error is noted in line 107 of Figure 3. (A, B) ….recurrent APLSA (authors to change to ALPSA) lesion (white arrow),….

Reply: Thanks for the comment. The typo is corrected.

References

  1. Park, S.S.; Loebenberg, M.L.; Rokito, A.S.; Zuckerman, J.D. The shoulder in baseball pitching: biomechanics and related injuries--Part 1. Bulletin of the NYU Hospital for Joint Diseases 2002, 61, 68-68.
  2. Skendzel, J.G.; Sekiya, J.K. Diagnosis and management of humeral head bone loss in shoulder instability. Am J Sports Med 2012, 40, 2633-2644, doi:10.1177/0363546512437314.
  3. Latarjet, M. [Treatment of recurrent dislocation of the shoulder]. Lyon Chir 1954, 49, 994-997.
  4. Park, J.-Y.; Lee, J.-H.; Oh, K.-S.; Chung, S.W.; Lim, J.-j.; Noh, Y.M. Return to play after arthroscopic treatment for shoulder instability in elite and professional baseball players. Journal of Shoulder and Elbow Surgery 2019, 28, 77-81.
  5. Rowe, C.R.; Zarins, B. Chronic unreduced dislocations of the shoulder. J Bone Joint Surg Am 1982, 64, 494-505.
  6. Williams, G.N.; Gangel, T.J.; Arciero, R.A.; Uhorchak, J.M.; Taylor, D.C. Comparison of the Single Assessment Numeric Evaluation method and two shoulder rating scales. Outcomes measures after shoulder surgery. Am J Sports Med 1999, 27, 214-221.
